Effect of operational parameters, characterization and antibacterial studies of green synthesis of silver nanoparticles using Tithonia diversifolia

Dada Adewumi O. 1 dada.oluwasogo@lmu.edu.ng
Inyinbor Adejumoke A. 1
Idu Ebiega I. 1
Bello Oluwasesan M. 2 3
Oluyori Abimbola P. 1
Adelani-Akande Tabitha A. 4
http://orcid.org/0000-0002-0621-5235 Okunola Abiodun A. 5
Dada Olarewaju 6
1 Industrial Chemistry Programme, Department of Physical Sciences, Nanotechnology Laboratory, Landmark University , Omu Aran, Kwara State , Nigeria
2 National Center for Natural Products Research, School of Pharmacy, The University of Mississippi , Oxford , MS , USA
3 Department of Applied Chemistry, Federal University Dutsin-Ma, Dutsin-Ma , Katsina State , Nigeria
4 Department of Biological Sciences, Microbiology Programme, Landmark University , Omu Aran, Kwara State , Nigeria
5 Department of Agric & Biosystem Engineering, Landmark University , Omu Aran, Kwara State , Nigeria
6 Nigerian Stored Product Research Institute (NSPRI) , Ilorin, Kwara State , Nigeria
Chang Huan-Tsung
Electronic publication date: 2018 Oct 30
Publication date: 2018
Volume: 6
Electronic Location ID: e5865
Received 2018 Apr 10; Accepted 2018 Oct 1
Copyright: © 2018 Dada et al.
Copyright year: 2018
Copyright holder: Dada et al.
License: This is an open access article distributed under the terms of the Creative Commons Attribution License, which permits unrestricted use, distribution, reproduction and adaptation in any medium and for any purpose provided that it is properly attributed. For attribution, the original author(s), title, publication source (PeerJ) and either DOI or URL of the article must be cited.
License URL: https://creativecommons.org/licenses/by/4.0/

Keywords: Green synthesis, Tithonia diversifolia, Operational parameters, FTIR, UV–Vis, Silver nanoparticles, Antimicrobial activity, TEM, SEM/EDX, XRD

Funding: The authors received no funding for this work.

==============================
Background

There is a growing interest in the green synthesis of silver nanoparticles (AgNPs) using plant extract because the technique is cost effective, eco-friendly and environmentally benign. This is phasing out the use of toxic and hazardous chemical earlier reported. Tithonia diversifolia is a wild sunflower that grows widely in the western part of Nigeria with a proven medicinal benefit. However, several studies carried out have left doubts on the basic operational parameters needed for the green synthesis of AgNPs. The objective of this work was to carry out green synthesis of AgNPs using T. diversifolia extract via an eco-friendly route through optimization of various operational parameters, characterization, and antimicrobial studies.

Method

Green synthesis of TD-AgNPs was done via bottom-up approach through wet chemistry technique using environmentally benign T. diversifolia plant extract as both reducing and stabilizing agent. Phytochemical Screening of the TD plant extract was carried out. Experimental optimization of various operational parameters—reaction time, concentration, volume ratio, and temperature was investigated. TD-AgNPs were characterized by UV–Vis spectroscopy, FTIR Spectroscopy, SEM/energy-dispersive X-ray spectroscopy (EDX), X-ray diffraction (XRD), and transmission electron microscopy (TEM). Antimicrobial studies against multi drug resistant microorganisms (MDRM) were studied using the agar well diffusion method.

Results

This study reveals the importance of various operational parameters in the synthesis of TD-AgNPs. Excellent surface plasmon resonance peaks (SPR) were obtained at optimum experimental factors of 90 min reaction time under room temperature at 0.001M concentration with the volume ratio of 1:9 (TD extract:Ag ion solution). The synthesis was monitored using UV–Vis and maximum wavelength obtained at 430 nm was due to SPR. The morphology and elemental constituents obtained by TEM, SEM, and EDX results revealed a spherical shape of AgNPs with prominent peak of Ag at 3.0 kV in EDX spectrum. The crystallinity nature was confirmed by XRD studies. FTIR analysis proved presence of biomolecules functioning as reducing, stabilizing, and capping agents. These biomolecules were confirmed to be flavonoid, triterpenes, and saponin from phytochemical screening. The antimicrobial studies of TD-AgNPs were tested against MDRM—Escherichia coli, Salmonella typhi, Salmonella enterica, and Bacillus subtilis.

Discussion

The variation of reaction time, temperature, concentration, and volume ratio played substantive and fundamental roles in the synthesis of TD-AgNPs. A good dispersion of small spherical size between 10 and 26 nm was confirmed by TEM and SEM. A dual action mechanism of anti-microbial effects was provided by TD-AgNPs which are bactericidal and membrane-disruption. Based on the antimicrobial activity, the synthesized TD-AgNPs could find good application in medicine, pharmaceutical, biotechnology, and food science.

Introduction

In the rapidly improving field of nanotechnology, nanomaterials are on the leading front. Their special property most especially the size gives them an edge over other materials. This improves their applications in various human activities (Subba Rao et al., 2013). Silver nanoparticles (AgNPs) among various metal nanoparticles have received significant consideration because they are effective antimicrobial agents that exhibit low toxicity; and have diverse in vitro and in vivo applications (Abou El-Nour et al., 2010). Organic and inorganic nanoparticles are the two broad group classifications of nanoparticles. AgNPs have been identified as peculiar inorganic nanoparticles due to its superior properties with functional versatility leading to unending interest among researchers (Shankar et al., 2004).

In this study, green synthesis approach has been adopted because it eliminates the use and generation of hazardous substances using a bio-friendly approach that is applicable to all parts of chemistry (Sharma, Chaudhary & Singh, 2008). Tithonia diversifolia plant is an ornamental shrub also known as Mexican sunflower native to Mexico and Central America from where it was introduced to Africa, Australia, Asia, and South America. It widely grows in Nigeria hence its common name, Wild Sunflower. It has several applications and diverse pharmacological applications. It possesses the following pharmacological properties: anti-inflammatory, analgesic, antinociception, antimalarial, antibacterial, antitumor, antidiabetic, antidiarrheal, antihelminthic, and antiviral properties (Kawlni et al., 2017). These properties of TD necessitated and gingered our research interest in utilizing it as ecofriendly and zero cost extract serving as bioreducing and stabilizing agent in the synthesis of AgNPs.

There are a number of studies on the green synthesis of AgNPs using different plant extracts. Syzygium aromaticum extract (Vijayaraghavan et al., 2012); Acalyphaindica leaf extract (Krishnaraj et al., 2010); Punica granatum peel extract (Edison & Sethuraman, 2013); banana peel extract (Ibrahim, 2015); Thevetia peruviana Juss (Oluwaniyi et al., 2015); Cavendish banana peel extract (Kokila, Ramesh & Geetha, 2015); Oak Fruit Hull (Jaft) extract (Heydari & Rashidipour, 2015); Artocarpus heterophyllus Lam. Seed Extract (Jagtap & Bapat, 2013) and Urtica dioica Linn. Leaves (Jyoti, Baunthiyal & Singh, 2016) were utilized in the green synthesis of AgNPs. Despite all these studies carried out, experimental optimization of operational parameters and factors influencing the synthesis of AgNPs have not been given a total consideration. More so, phytochemical screening of T. diversifolia leaves extract, experimental optimization of operational parameters in the green synthesis of T. diversifolia silver nanoparticles (TD-AgNPs), the characterization and application of TD-AgNPs have not been reported hence the need for this study. Furthermore, application of T. diversifolia biosynthesized silver nanoparticles (TD-AgNPs) on multi-drug resistance micro-organisms such as Escherichia coli, Salmonella typhi, Salmonella enterica, Bacillus subtilis has not been reported. These multi-drug resistance microorganisms (MDRM) are grouped as Gram-positive and Gram-negative bacteria. Gram-positive bacteria give a positive test in Gram stain test; they have peptidoglycan layers, produce primarily exotoxins, high resistance to physical disruption, high susceptibility to anionic detergent, and resistance to drying. However, Gram-negative bacteria are negative to Gram stain test, they have single peptidoglycan layer with periplasmic space. They have low resistance to physical disruption, low susceptibility to anionic detergents, and well as resistance to drying. Compared with Gram-positive bacteria, Gram-negative bacteria are more resistant against antibodies because of their impenetrable cell wall. They are more virulent than Gram-positive bacteria (Hoerr et al., 2012; Ramachandran, 2014). Hence, the main reason for the choice of three Gram-negative bacteria and one Gram-positive. The aims of this study are to: investigate the phytochemical screening of T. diversifolia leaves extract; experimentally optimized various factors influencing the operational parameters in the green synthesis of T. diversifolia silver nanoparticles (TD-AgNPs); carry out characterization and application of T. diversifolia silver nanoparticles (TD-AgNPs) on MDRM.

Materials and Methods

Collection of TD leaves, preparation of T. diversifolia extract and phytochemical screening

Tithonia diversifolia plant (Fig. 1) was collected in Landmark University vicinity, slightly washed in order to remove the farm land soil and air-dried to avoid losing vital volatile molecules. The dried leaves were pulverized and 10 g of fine power of TD was added to 500 mL deionized water at 100 °C and left for 10 min. The extract was filtered using Whatman 185 μm filter paper. Phytochemical screening was carried out to identify the presence of phenols, saponins, triterpenes, flavonoids, alkaloids, and steroids in the TD leaf extract. These various tests were done following the procedure in the literature (Dada, Adekola & Odebunmi, 2015; Senguttuvan, Paulsamy & Karthika, 2014).

Figure 1 A typical T. diversifolia plant.

Source credit: Ebiega I Idu.

Synthesis of TD-AgNPs and experimental optimization of operational parameters

In a typical procedure, 10 mL of the leaf extract was measured and poured into a clean 250 mL beaker and reacted with 90 mL of 1 × 10−3M AgNO3 at room temperature. The resulting solution was stirred on the mechanical shaker at optimum operational conditions. T. diversifolia silver nanoparticles (TD-AgNPs) formed was separated by centrifugation at 4,000 rpm for 10–15 min.

Experimental optimization of operational parameters

Effects of four important operational parameters (experimental factors) which are concentration, reaction time, volume ratio, and temperature on the formation of TD-AgNPs were investigated and the study was monitored using Biochrom Libra PCB 1500 UV–VIS spectrophotometer. Detail on the procedure has been provided in the supplementary material of this article (S1). The investigation was carried out specifically optimizing the concentrations of Ag+ solution (0.001–0.01M); reaction time from 5 to 90 min; Volume ratio of plant to Ag+ solution in the ratio 1:9, 2:8, 3:7, 4:6, 5:5, 6:4, 7:3, 8:2, 9:1, and effect of temperature.

Characterization of TD-AgNPs

All operational factors studied were monitored using Double beam Biochrom Libra PCB 1500 UV–VIS spectrophotometer. FTIR analysis was done for the determination of functional groups present in leaves extract of T. diversifolia responsible for the formation of Ag nanoparticles that was actualized using SHIMADZU FTIR model IR8400s spectrophotometer. Energy-dispersive X-ray spectroscopy (EDX) profile coupled with the morphology determination via SEM was carried out using a TESCAN Vega TS 5136LM SEM typically at 20 kV at a working distance of 20 mm. Transmission electron microscopy (TEM) analysis was on Zeiss Libra 120 @ 80 kV.

Results

Phytochemical screening

Qualitative phytochemical screening analysis was done on T. diversifolia leaf extract to determine the presence of some phytochemicals presence in the leaves of this medicinal plants used. The result represented in Table 1 indicates the presence of Saponins, triterpenes, flavonoid, and steroids confirming the availability of polyols which serve as the stabilizing and reducing agent. This result obtained is corroborated in the literature (Pochapski et al., 2011). Detail of the phytochemical screening test are presented in the supplementary material.

Table 1 Phytochemical screening test result on T. diversifolia.

S/N	Phytochemical screening test done	Tithonia diversifolia leaf extract	
1.	Test for phenol (FeCl3 test)	−	
2.	Test for Saponins (Froth’s test)	+	
3.	Test for triterpenes	+	
4.	Test for flavonoids		
	 (a) Alkali’s test	+	
 (b) Lead acetate test	+	
5.	Test for alkaloids	−	
 (a) Mayer’s test		
6.	Test for steroids (Salkowski’s test)	+	
7.	Test for sterols (Libermann–Buchard)	−	
Notes:

Table key: +, present; −, absent.

Effects of operational parameters

The synthesis of AgNPs depends largely on some operational parameters. These are factors that influence nanoparticles synthesis irrespective of the technique used. In this study, evaluation of several important experimental factors, including reaction time from 5 to 90 min, temperature, concentration of 0.001, 0.002, 0.004, 0.006, 0.008, and 0.01M silver ion solution and volume ratio of 1:9, 2:8, 3:7, 4:6, 5:5, 6:4, 7:3, 8:2, 9:1 (silver ion solution to TD extract) were studied. Each of these experimental factors was monitored by UV–Vis spectroscopic measurements.

Effect of reaction time

The reaction time and the temperature operational parameters play substantive roles in the synthesis TD-AgNPs. The effect of reaction time was investigated by steady monitoring the reaction of the plant extract and AgNO3 for 5, 10, 20, 30, 45, 60, and 90 min at room temperature. The moment TD extract reacts with the solution of AgNO3, a color change was observed from green to brown within 10 min of reaction. The color intensified with increase in time (Balavandy et al., 2014). UV–Vis measurements were taken at various time intervals as shown in Fig. 2A. It can be inferred that between zero and 10 min, the surface plasmon resonance (SPR) band is broadened because of the slow conversion of silver ion (Ag+) to zerovalent silver (Ag0) nanoparticles. Excellent SPR band was observed as the reaction time increases because large amount of Ag+ has been converted to Ag0. The UV–Vis spectra measured showed the absorption of TD-AgNPs synthesized nanostructures and best SPR peak was observed within 430 nm at 90 min. Reports from the literature have shown that when the color is stable and a narrow shape of the SPR has been achieved, optimum time is reached. Supporting this observation is the outcome of the study by Mohamed et al. (2014) and Anandalakshmi, Venugobal & Ramasamy (2016) where a rapid synthesis was obtained at lower time and this was their optimum time. The UV–Vis spectra measured showed the absorption of TD-AgNPs synthesized nanostructures and best SPR peak was observed within 430 nm at 90 min. Further investigation of other operational parameters was carried out at 90 min which is the optimum time obtained.

Figure 2 Effects of operational parameters (resubmission).

UV–Vis absorption spectra for experimental optimization on: (A) effect of contact time, (B) effect of temperature at 45 °C, (C) effect of temperature at 55 °C, (D) effect of concentration, (E) effect of volume ratio.

Effect of temperature

A further study on the effect of temperature on the synthesis of AgNP was carried out at 45 and 55 °C as shown in Figs. 2B–2C. From the literature, it has been reported that increase in temperature leads to increase in the intensity of the SPR band as a result of bathochromic shift resulting in a decrease in the mean diameter of AgNP (Bindhu & Umadevi, 2014). This however, may not connote the optimum temperature where excellent SPR band maybe obtained. In this study, excellent representation was obtained at room temperature because the biomolecules from the TD extract effectively reduced and stabilized AgNPs at ambient temperature. Stable TD-AgNPs was formed at room temperature thus justifying the green synthetic route.

Effect of concentration

Depicted in Fig. 2D is the UV–Vis spectra of effect of concentration on the synthesis of TD-AgNPs. This operational parameter was monitored at various concentrations of silver ion solution and at optimum conditions. The investigation was carried out on the following concentration: 0.001, 0.002, 0.004, 0.006, 0.008, and 0.01M. The intensity increases as the concentration of Ag+ increases with the SPR peak for all the different concentrations. A distinctive SPR peak at 430 nm was obtained at 0.001M Ag+ concentration. Varying the concentration of Ag+ solution affects the size and shape of the AgNPs (Filippo et al., 2010).

Effect of volume ratio

Portrayed in Fig. 2E is the surface plasmon peaks on the investigation of effect of volume ratio. This was studied varying the volume ratio of the leaf extract to 0.001M Ag+ solution in the ratio 1:9, 2:8, 3:7, 4:6, 5:5, 6:4, 7:3, 8:2, 9:1. The absorption peaks were broader and irregular at higher volume of extract indicating a slow reduction of Ag+ to Ag0 and presence of AgNPs with broader size distribution (Peng, Yang & Xiong, 2013; Oluwaniyi et al., 2015). As the volume of Ag+ solution increased, the absorption peak became sharper with excellent enhancement in the absorption band intensity at 430 nm. The SPR peaks in UV–Vis spectra showed best representation in ratio 1:9 (TD extract:Ag+ solution). This indicates that TD extract stabilizes and bioreduces silver ion at ratio 1:9 giving 430 nm as a result of SPR. Thus further study was carried out using the optimum volume ratio.

Characterization

UV–Vis spectroscopic study

The most imperative characterization technique for studying the synthesis of AgNP is the UV–Vis spectroscopy. In this study, the color change was observed from the absorption in the visible range. The absorption of light occurs in the visible region of the electromagnetic spectrum where atoms and molecules undergo electronic transition of π-π*, n-π*, σ-σ*, and n-σ*. Absorption of energy in the form of ultraviolet or visible light is by molecules containing π-electrons or non-bonding electrons (n-electrons) to excite these electrons to higher anti-bonding molecular orbitals. The length of wave depends on the excitation of the electrons, the more easily excited the electrons, the longer the wavelength of light it can absorb (Dada et al., 2018). Oscillation of electron at the surface of AgNPs brought about the SPR resulting from the change of color from green to yellow and finally brown. UV–V is measurements were taken to study the formation of silver nanostructures in the reaction of T. diversifolia with silver nitrate (AgNO3) and this is presented in Fig. 3A.

Figure 3 Characterization of TD-AgNPs.

(A) UV–Vis absorption spectrum, (B) FTIR spectrum, (C) SEM image, (D) EDX spectrum, (E) TEM image, and (F) XRD pattern of TD-AgNPs.

FTIR spectroscopic study

The result of the phytochemical screening was corroborated by the FTIR spectroscopic study. Presented in Fig. 3B is the FTIR result of TD-AgNps identifying the biomolecules that were bound specifically on the TD-AgNPs. It is obvious that the biomolecules are responsible for the reduction of Ag+ to Ag0. This was well elucidated in the Discussion section of this article.

SEM, EDX, TEM, and XRD studies

Important characterization signatures were provided by SEM, EDX, TEM, and XRD results which are very imperative to this study.

SEM identifies the surface characteristics, morphology and the distribution of the TD-AgNPs depicted on the SEM micrograph (Fig. 3C; Dada, Adekola & Odebunmi, 2017a).

Energy-dispersive X-ray spectroscopy gives information on the surface atomic distribution and the chemical elemental composition of metallic nanoparticles. Figure 3D depicts the EDX of TD-AgNPs which reveals a very strong signal in the silver region at three kV and confirms the formation of AgNPs.

The TEM is also one of the valuable tools for characterization of metallic nanoparticles because it unravels the size, shape and morphology. Depicted in Fig. 3E is the TEM image of TD-AgNPs showing a characteristic spherical shape of Ag nanoparticles.

X-ray diffraction result revealed the crystalline structure of TD-AgNPs as shown in Fig. 3F. Four distinct characteristic peaks indicated at angles 38°, 44°, 65°, and 78°.

Antimicrobial Studies

The antimicrobial study was carried out using agar well diffusion method. 0.2 mL of the TD-AgNPs solution, TD leaf extracts, the positive control (Ciproflaxcin), and negative control (sterile water) were introduced into the well accordingly. The plates were left to diffuse for 1 h before placing them in an incubator at 37 °C for 24 h. After the incubation period, the mean diameters of the zones of inhibition around the wells were recorded and presented in Table S1. The results of the antimicrobial studies are presented in Fig. S1, Fig. 4 and Table S2. Shown in Fig. S1 are the plates of the various zones of inhibitions for different bacteria investigated. The measurement of the zone of inhibition is presented in Table S1. However, Fig. 4 showed the bar chart representation of the antimicrobial activity of synthesized silver nanoparticles (TD-AgNPs), TD Extract, Positive Control, and Negative Control against Escherichia coli, Salmonella typhirium, Salmonella enterica, and B. subtilis. The result indicated TD-AgNPs is very effective against these multi-drug resistance organisms while both the TD leaves extract and the negative control sample was not active at all.

Figure 4 Antimicrobial activity of synthesized silver nanoparticles (TD-AgNPs).

Antimicrobial activity of synthesized silver nanoparticles (TD-AgNPs), TD extract, positive control and negative control against E. coli, S. typhirium, S. enterica, and Bacillus.

Discussion

The aims of this study were successfully achieved. Phytochemical screening revealed the presence of functional biomolecules responsible for the bioreduction of Ag+ to Ag0. This study has examined four major operational parameters as revealed in Figs. 2A–2E. These are imperative to the synthesis of AgNPs. The operational parameters were monitored using the UV–Vis spectrophotometer. The study established that excellent SPR peaks formed at 430 nm were obtained at reaction time of 90 min (Fig. 2A), under optimum experimental conditions. Effect of temperature at 45 °C (Fig. 2B) and 55 °C (Fig. 2C) revealed the dependence of the TD-AgNPs synthesis on temperature. However, the room temperature synthesis is greener than the heated syntheses, which is a further advantage. The effect of concentration affects the size of the TD-AgNPs. At higher concentrations (0.004; 0.006; 0.008, and 0.01M), there was change in the intensity as a result of bathochromic shift leading to broad band, lower size, dispersion, and higher aggregation. However, at lower Ag+ concentrations (0.001 and 0.002M), higher intensity, better absorbance and narrower bands were observed as seen in Fig. 2D. The effect of concentration resultantly influences its particle size. SPR band maximum intensity and band width are influenced by particle shape, dielectric constant of the medium, and temperature (Narayanan & Sakthivel, 2011). This enhanced a good shape and size control. This finding is supported by the report of Kokila, Ramesh & Geetha (2015). Best SPR was obtained at 0.001M concentration which gives a well dispersed size ranging between 10 and 26 nm with a spherical characteristics shape confirmed by TEM and SEM. Best volume ratio of 1:9 (TD extract:Ag+ solution) was observed suitable for better and stable TD-AgNPs formation.

TD-AgNPs were characterized by UV–Vis (Fig. 3A), FTIR (Fig. 3B), SEM (Fig. 3C), EDX (Fig. 3D), and XRD (Fig. 3E). Figure 3A revealed that the maximum absorption was observed at 430 nm which was due to the AgNPs SPR band. The SPR is as a result of the free electron arising from the conduction and valence bands lying close to each other in metal nanoparticles (Anandalakshmi, Venugobal & Ramasamy, 2016; Dada et al., 2018). This SPR peak gives a convenient spectroscopic signature for the formation of AgNP and a clue on the spherical shape of AgNP. This corroborates with the TEM measurement (Pandey, Goswami & Nanda, 2012; Van Dong et al., 2012).

The FTIR spectrum was recorded in the region of 4,000–500 cm−1 region (Fig. 3B) signifying the absorbance bands centered as follows: 3,321 cm−1 is assigned to polyols; 2,240 cm−1 corresponds to C–H stretching vibration; peak at 1,692 cm−1 to N–H vibration stretching; peak at 1,615 cm−1 corresponds to –C=C– of aromatic ring; 1,555 cm−1: C–N stretching of amines; 1,194 cm−1 for C–N stretching of aromatic amine group and the bands observed at 1,009 cm−1 corresponds to C–H stretching of polysaccharides; 665 cm−1: N–H wag of amines. FTIR result obtained confirmed the phytochemical screening result of some biomolecules. It implies that the biomolecules functioned as reducing, capping, and stabilizing agents. Analysis of FTIR result indicates that the AgNPs were surrounded by terpenoids, alcohols, lactone, and carbonyl group from amine serving as strong binding site for AgNPs (Dubey, Lahtinen & Sillanpää, 2010; Edison & Sethuraman, 2013; Tran, Vu & Nguyen, 2013; Dada, Adekola & Odebunmi, 2017b).

It is evident from the SEM micrograph (Fig. 3C) that the morphology of TD-AgNP is spherical and this is in good agreement with the shape of SPR band in the UV–Vis spectrum (Benn & Westerhoff, 2008; Singh, Saikia & Buragohain, 2013; Dada, Adekola & Odebunmi, 2017c).

Figure 3D depicts the EDX spectrum of TD-AgNPs which reveals a very strong signal in the silver region and confirms the formation of AgNPs. Metallic silver nanocrystals have a characteristic peak at three kV due to SPR. The other peaks observed were found to be other elemental constituents in the plants and the gold (Au) seen on the spectrum resulted from the preparation of the samples for the EDX characterization (Bankura et al., 2012; Seo et al., 2012; Dada, Adekola & Odebunmi, 2015).

The characteristic spherical shape of TD-AgNPs is further confirmed from the TEM image presented in Fig. 3E. A good dispersion of small spherical size between 10 and 26 nm was observed (Babu & Prabu, 2011; Prathna et al., 2011). The antimicrobial activity is a function of the size of the nanoparticle (Tippayawat et al., 2016).

Depicted in Fig. 3F is the X-ray diffraction result which confirmed the crystalline structure of TD-AgNPs. The four intense peaks appearing around 38°, 44°, 65°, and 78° fits in perfectly to the (111), (200), (220), and (311) lattice planes. This maybe indexed as the band for face centered cubic structures of silver. This XRD result confirmed the crystallinity nature of AgNPs synthesized using T. difersifilia extract (Bar et al., 2009; Wen et al., 2012).

The in vitro antimicrobial studies on MDRM were carried out using leaf extract of T. diversifolia, TD-AgNPs synthesized, sterile water (negative control), and Ciproflaxcin (Positive control and for comparison of the effectiveness of synthesized TD-AgNPs). Details on the antimicrobial procedure are stated in the supplementary material of this article (S2–S8) and Table S2. Figure 4 shows the result of the antimicrobial activity indicating the growth inhibition of the TD-AgNPs and the positive control. The results showed that the leaf extracts of T. diversifolia and the negative control (sterile water) had no significant activity or effect on the microorganisms. This finding is supported by the study carried out by Tran, Vu & Nguyen (2013). However, the significant inhibitory antimicrobial activity was shown by synthesized silver nanoparticles (TD-AgNPs) with inhibition zones varying from 10 to 15 mm (Table S1). These results were further analyzed statistically to compare the inhibitory effect of TD-AgNPS to the positive control (Ciproflaxcin) used as shown in Fig. 4. More inhibitory activity of the synthesized nanoparticles occurred on Bacillus subtilis with inhibition zone of 15 ± 0.34 mm than the rest of the microorganisms as observed. It was also observed in relative terms versus the positive control, the best inhibition seems to be S. enterica. A dual action mechanism of anti-microbial effects was provided by TD-AgNPs which are bactericidal and membrane-disruption. This is corroborated by the report of Jain et al. (2009); Sharma, Yngard & Lin (2009).

Conclusion

The green synthesis of AgNP using eco-friendly and environmentally benign T. diversifolia plant extract was successfully carried out. This study shows that the synthesis of T. diversifolia silver nanoparticles (TD-AgNPs) depends on various experimental operational parameters. It can be concluded that optimum concentration of 0.001M Ag+ solution, reaction time of 90 min, ambient temperature for stability of biomolecules, and volume ratio of 1:9 favors the optimum yield of TD-AgNPs. TD-AgNPs were characterized by different spectroscopic and microscopic techniques. The presence of biomolecules (flavonoids and terpenoids) in TD extract observed from the phytochemical screening was confirmed by FTIR spectroscopic study. These biomolecules serve as the reducing, stabilizing, and capping agents changing Ag+ to Ag0. Surface plasmon peak was observed at 430 nm by UV–Vis spectroscopic measurement. Spherical shape and 10–26 nm size of TD-AgNPs were determined by SEM and TEM. Elemental composition of TD-AgNPs with an intense peak of Ag at 3.0 kV was determined by EDX and the crystallinity nature of Ag nanoparticles by XRD. Antimicrobial studies carried out against multidrug resistance microorganism showed the efficacy and efficiency of TD-AgNPs as observed in the inhibitory function. It is obvious that TD-AgNPs showed activity against Gram-positive and Gram-negative micro-organism. It can therefore be concluded that TD-AgNPs would find application in medicine, pharmacology, and food science.

Supplemental Information

Supplemental Information 1 Experimental procedures for Operational Parameters, Antimicrobial Studies and Raw Results of Characterization.

Click here for additional data file.

Supplemental Information 2 The plates showing the various zones of inhibitions for different bacteria. From left E.coli, Bacillus, Salmonella typhi and Salmonella enterica.

Click here for additional data file.

Supplemental Information 3 Raw data for the phytochemical screening test (+/− sign is not error analysis rather indication of presence or absence in the screening test).

Click here for additional data file.

Supplemental Information 4 Raw data of Figure 4.

Click here for additional data file.

The authors appreciate the Management of Landmark University for providing research facilities and an enabling environment for result-oriented studies for breaking new grounds.

Additional Information and Declarations

Competing Interests

Author Contributions

Data Availability

The authors declare there are no competing interests.

Adewumi O. Dada conceived and designed the experiments, performed the experiments, analyzed the data, contributed reagents/materials/analysis tools, prepared figures and/or tables, authored or reviewed drafts of the paper, approved the final draft, corresponding author and initiator of the research.

Adejumoke A. Inyinbor conceived and designed the experiments, performed the experiments, analyzed the data, contributed reagents/materials/analysis tools, prepared figures and/or tables, authored or reviewed drafts of the paper, approved the final draft.

Ebiega I. Idu conceived and designed the experiments, performed the experiments, analyzed the data, contributed reagents/materials/analysis tools, prepared figures and/or tables, authored or reviewed drafts of the paper, approved the final draft.

Oluwasesan M. Bello performed the experiments, analyzed the data, contributed reagents/materials/analysis tools, prepared figures and/or tables, authored or reviewed drafts of the paper, approved the final draft, significant views towards the work.

Abimbola P. Oluyori performed the experiments, analyzed the data, contributed reagents/materials/analysis tools, prepared figures and/or tables, authored or reviewed drafts of the paper, approved the final draft.

Tabitha A. Adelani-Akande performed the experiments, analyzed the data, contributed reagents/materials/analysis tools, prepared figures and/or tables, authored or reviewed drafts of the paper, approved the final draft.

Abiodun A. Okunola performed the experiments, analyzed the data, contributed reagents/materials/analysis tools, prepared figures and/or tables, authored or reviewed drafts of the paper, approved the final draft, significant encouragement.

Olarewaju Dada performed the experiments, analyzed the data, contributed reagents/materials/analysis tools, prepared figures and/or tables, authored or reviewed drafts of the paper, approved the final draft, significant opportunities and applications in green synthesis.

The following information was supplied regarding data availability:

The raw data are provided in the Supplemental Files.

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
