# Peer review of "Effect of operational parameters, characterization and antibacterial studies of green synthesis of silver nanoparticles using Tithonia diversifolia"

_PeerJ, doi:10.7717/peerj.5865_

## Round 0.1 · original submission · Major Revisions

The authors are encouraged to resubmit their manuscript after careful revision according to the reviewers' comments - primarily reviewer 1 and 3.

Reviewer 1 ·

Basic reporting

- The authors should carefully proofread the manuscript for English mistakes. I have not listed them all here but will point out a few.
- Some expressions such as “early in vogue” are ambiguous (line 27). I understand the appeal of creative writing, but it may confuse some readers.
- Line 27. “facing out” should be “phasing out”.
- Line 52, 221, 285, 298, Fig 4 (image and caption) says only “Bacillus” but this should be “Bacillus subtilis”
- The authors say 430 nm on lines 156 173, 182, 184, 242 but say 435 nm on lines 45, 45, 314. I think this should be consistent.
- Lines 79-81. These pharmacological properties of the flower should be cited.
- Lines 69, 283. In vitro and in vivo should be in italics.
- Throughout the manuscript these bacteria are called multi-drug resistant. I am not a specialist about bacteria, but I am wondering if these are multi-drug resistant species (e.g. MRSA). If I am wrong here, please disregard.
- Line 310. This should say “electron microscopic techniques”.
- Lines 319-320. The authors could note which bacteria are Gram Positive and which are Gram Negative. A small explanation (in the Discussion or maybe Introduction) about the difference could also be added elsewhere. This information may be helpful for readers whose focus is nanomaterials but not bacteria.
- Figure 2B. The IR peaks are hard to read, at least in the main text (the supplementary material one is fine). Perhaps the authors could make better labels for the important peaks that they mention in the text.

Experimental design

- Line 105. The cleaning process should be explained.
- Line 108. The phytochemical screening processes should be briefly explained here or elsewhere (Line 133, or lines 234-235) in the paper for the reader to understand. Many people interested in nanoparticle synthesis will have never heard of these tests.
- Line 114. Was the solution stirred or shaken? The supplementary material mentions “Agitating” but it is unclear what this means.
- For the antibacterial studies, it is not clear if the same amount of pure extract was used as is approximately present in the TD-AgNPs. I realise that this may be difficult to know exactly. If this authors have taken this into consideration, it should be mentioned somewhere.

Validity of the findings

- Lines 139-143. This is just a re-summary of the method. The results should be stated here.
- Line 158. The authors claim that 90 min is the optimum synthesis time. It certainly is better than the shorter time periods. Why did they not try a longer synthesis time? 90 min may not be the true optimum. If possible, the authors could examine longer synthesis time. If not, it could at least be discussed.
- Line 159. The authors could explain how these other studies support their findings.
- Line 161. “From the literature”… but there is no citation. Please cite whatever you have in mind.
- Lines 173-174. How does varying the concentration affect size and shape? This could be added here or in the Discussion.
- Line 180. The effect of UV-vis peak shape to nanoparticles with a broader size distribution is known among the nanoparticle community but may not be known to all readers. Accordingly, the authors should add a citation here.
- Line 193. The results should be described here. Again, it just seems like a repetition of the methods rather than a proper Results section.
- Lines 236-244. The authors could note that the room temperature synthesis is greener than the heated syntheses, which is a further advantage.
- Lines 247-248. This needs a reference.
- Lines 251-253. The authors should elaborate on the relation between the nanoparticle shape, SPR and TEM. They provide references, but do not actually explain how their data relate to these findings. This will improve their paper since it will be informative, rather than forcing the reader to look elsewhere.
- Lines 255-256. The rest of the FTIR seems fine, but is the 2240 cm-1 peak from C-H stretching? I am not an IR expert, but I am not aware of the C-H vibration here. The authors could improve the paper by providing a citation if this is truly the assignment.
- Lines 261-262. The authors could further explain their points here to make the paper better (similarly to the concern above about the nanoparticle shape, SPR and TEM). Again, what they have written is not wrong, but it is vague and makes the paper less useful (i.e. citable) for readers. I encourage the authors to go into greater detail to improve their paper.
- Line 266. The AgNPs look more round like a ball-shape than a rod-shape to me, especially in the TEM.
- Line 296. The authors note that the results were “analyzed statistically” and mention the standard deviation in one case (line 299). Error bars for should be added to Figure 4 to improve the graph.
- The authors note the TD-AgNPs work best against Bacillus. This is a useful finding. I also noticed however that in relative terms vs the positive control, the best case seems to be S. enterica. While not mandatory, this is something I would add to the manuscript.

Comments for the author

Overall I have a good impression of this paper. I have noted many issues here, but my intention is simply for it to be improved. The issues are more with the manuscript and interpretation of the data, rather than problems with the experiments. My opinion is that this article is scientifically valid, and it could be accepted for publication in PeerJ after consideration of the various points I have noted above. These improvements as a major revision will provide further insight for the reader, and perhaps make the paper more citable.

Reviewer 2 ·

Basic reporting

There are too many grammer errors and odd sentences. The Figures are also not professional.

Experimental design

no comment.

Validity of the findings

Since 2008, there have been too many reports to synthesize the AgNPs using different plant extracts and microorganisms, including optimization conditions, characterization, and antibacterial test. Actually, researchers gradually concluded that most of the plants extracts can mediate the production of AgNPs at special conditions. The novelty and impact are not obvious for this manuscript, at least far from the standard of this journal.

Comments for the author

The language should be revised completely, and the data such as UV-vis figures should be more professional. The main problem is lacking the obvious novelty and advanced impact. The experiment design is complete, while it can't reach the standard of PeerJ.

Reviewer 3 ·

Basic reporting

There are numerous grammatical errors throughout the manuscript - examples are but not limited to….

1. vis-à-vis is not a scientific word and should not be used.
2. Line 126 - ‘used’ is wrong tense
3. Line 141 - ‘The colour intensified with increase in time’ no article before increase IE ‘with an increase’
4. line 106 - mL or ml
5. Formatting 1 or 2 spaces after a a period - be consistent.
6. SPR is defined multiple times after first use.
7. line 279-81 ‘It was observed that more inhibitory activity of the synthesized nanoparticles occurred more on Bacillus with inhibition zone of 15 34 mm than the rest of the microorganisms.’
8. line 224 - nitty-gritty is not a scientific word normally used.

A great deal of the references are greater than 5 years old, I find it difficult to believe there are not some relevant update references in this area.

Experimental design

This research fits into the aims and scope of the journal.

The gap in the research literature is identified and defined, the meaningfulness and relevance in the gap is not fully explained however.

For the experimental design the methods are described in sufficient detail to be reproduced and seem of a high technical and ethical standard.

Validity of the findings

The data is robust, and apart from fig 4 which has no error bars.

Figure 3 - (b) FTIR spectra is not analyzed for specific peaks relevant to the text or work

Conclusions seem to fit with original research question.

---

## Round 0.2 · Minor Revisions

Although the authors have responded most of the comments from the reviewers carefully, many typos are still found. Description/discussion of the results shall be more precisely and specific, which will make it easier to follow. English polish is required.

---

## Round 0.3 · Minor Revisions

Although preparation of Ag NPs from silver ions in the presence of plant extracts is not new, the result shows potential of Ag NPs as antibacterial reagents. The manuscript must be polished and edited more carefully.

For example, TEM image can not provide the information of Ag element, meanwhile EDX can not reveal the size of Au NPs. Thus the sentence needs to be divided to two.

On the same page, contact time is improper; The sentence can be rewritten like: We evaluated several important experimental factors, including reaction time and temperature, concentration of what and volume ratio of what. Concentration and volume ratio must be specified.

---

## Round 0.4 · accepted · Accept

The authors have responded to my comments carefully. The revised manuscript provides some useful information for green preparation of silver nanoparticles that show antibacterial activity.

#